


# Tropical drought risk: estimates combining gridded vulnerability and hazard data

Alexandra Nauditt[1], Kerstin Stahl[2], Erasmo Rodríguez[3], Christian Birkel[4], Rosa Maria Formiga-Johnsson[5], Kallio Marko[6], Hamish Hann[1], Lars Ribbe[1], Oscar M. Baez-Villanueva[1,7], Joschka Thurner[1]

[1] Institute for Technology and Resources Management in the Tropics and Subtropics, Cologne Technical University of Applied Sciences, Germany
[2] Chair of Environmental Hydrological Systems, University of Freiburg, Germany
[3] Civil and Agricultural Engineering Department, Universidad Nacional de Colombia – Bogotá, Colombia
[4] Department of Geography, University of Costa Rica
[5] Department of Environmental and Sanitary Engineering, State University of Rio de Janeiro (UERJ), Brazil
[6] Department of Built Environment, Aalto University, Finland
[7] Faculty of Spatial Planning, TU Dortmund, Germany

*Correspondence to*: Alexandra Nauditt, alexandra.nauditt@th-koeln.de

**Abstract.**

Droughts are causing severe damages to water abundant tropical countries worldwide. Their resilience to water shortages tends to be low, often due to a lack of water infrastructure. Moreover, drought characteristics and risk in tropical catchments are poorly understood, which makes it difficult to select adequate adaptation measures. Thus, reliable methodologies to evaluate spatially distributed drought risk in data scarce tropical catchments are urgently needed.

We combined drought hazard and vulnerability related information to assess drought risk in four rural tropical study regions, the Muriaé, subcatchment of the Paraíba do Sul in Southeast Brazil, the Tempisque-Bebedero basin in North Costa Rica, the upper part of the Magdalena basin, Colombia and the Srepok, a Mekong tributary shared by Cambodia and Vietnam. Drought hazard was defined based on three variables, daily river discharge and precipitation and vegetation condition. Conditions below defined thresholds were transformed into a cumulative drought index. To assess vulnerability, we reclassified and weighted globally and regionally available gridded socioeconomic data to represent the potential of a drought to cause damages in selected socioeconomic sectors of rural tropical regions. Besides illustrating the relative severity of each indicator value, we developed drought risk maps combining hazard and vulnerability severity for each grid cell.

While for the Muriaé our results clearly identified the downstream area as being exposed to severe drought risk, the Tempisque showed highest risk along the major streams and related irrigation systems. Risk hotspots in the Upper Magdalena were found in the central valley and the dryer Southeast and in the Srepok



in the agricultural areas of Vietnam and downstream in Cambodia. Plausibility of results was confirmed by
local scientists and stakeholders, who evaluated the results for each indicator and risk hotspot. The
presented risk assessment methodology for data scarce and rural tropical areas offers a holistic, science
based and innovative solution to provide relevant drought related information. Being applied to individual
catchments, the findings described in this article will enable the selection of data sets, indices and their
classification - depending on basin size, spatial resolution and seasonality. At its current stage, the outcomes
of this study provide relevant information for regional planners and water managers dealing with the control
of future drought disasters in tropical regions.

## 1    Introduction

Droughts are a recurrent phenomenon in tropical regions worldwide (Adamson and Bird, 2010; Erfanian et
al., 2017) and are expected to become more severe in the future (Sheffield et al., 2018). Recent drought
disasters occurred in South-East Brazil, January 2014 to December 2015 (Nauditt et al., 2019b; Ribbe et al.,
2018), followed by an El Niño triggered drought in 2015-2016 affecting Costa Rica (Herrera and Ault, 2017),
Colombia (FAO, 2017) and Southeast Asia (Thirumalai et al., 2017), with devastating implications for
domestic water supply, agricultural and hydropower production, navigation, fire occurrence and public health
(Hoyos et al., 2017). Rural areas that rely on rain-fed agriculture, livestock and milk production, were
extremely impacted due to the lack of water storage or distribution infrastructure (Nauditt et al., 2019a). To
avoid such economic losses during future drought events, the respective governments have been seeking
for more effective adaptation strategies (IDEAM et al., 2014; Emater-RIO et al. 2016; FAO, 2017; UNGRD
et al., 2018). Decisions related to drought adaptation, though, need to rely on a profound knowledge about
drought hazard, vulnerability and exposure; spatially varying risk information that is rarely available in data
scarce tropical regions.

Many concepts have been developed to evaluate drought risk (Carrão et al., 2016; Stahl et al., 2016; Vogt
et al., 2018; Naumann et al., 2019; Meza et al., 2019), varying in their definition and interpretation of the
terms "risk", "hazard", "vulnerability" and "exposure" (González-Tánago et al., 2015). Nonetheless, although
varying in terminology, there is a wide agreement that risk cannot be understood by looking only at either
climate anomalies or only at socioeconomic vulnerability factors (UN-ISDR, 2009; Bachmair et al., 2017).
Understanding risk requires a more holistic evaluation of different conditions leading to drought disasters:
What extent and duration of a hydro-climatic deficiency caused drought impacts at which location? How did
the catchment, vegetation and discharge respond to this extreme event? Which environmental and economic





sectors were affected? Since such characteristics are climate, region- and sector specific, there is a demand
to design locally suitable drought risk assessment approaches and related data sets (Naumann et al., 2019).

The scale of analysis matters. Widely applied monthly scale standardized indices (eg. SPI 12) are useful for
large scale drought risk assessment (Naumann et al., 2018; Vogt et al., 2018). Tropical climates are often
dominated by a strong seasonality and a topography-influenced spatial rainfall variability. Few days without
rainfall might lead to a severe precipitation deficit that can affect cattle grazing and rain-fed agricultural
production. Indices based on monthly hydro-meteorological values might not detect short-term deficits in
quickly responding catchments. For tropical regions, it has therefore been proven useful to assess
meteorological and hydrological drought hazard at a daily timescale (Nauditt et al., 2017; Firoz et al., 2018).
Also the spatial distribution and coverage of hydro-climatic observations used to detect drought anomalies
are of key importance for hazard assessment. During drought, topography, geology, soil and land-cover
catchment characteristics as well as human water interventions influence hydrological processes, catchment
storage and release and therefore play a major role in the evolution of low flows (Bruijnzeel, 2004; Calder et
al., 2007; Birkel et al., 2012; Stoelzle et al., 2014; van Loon and Laaha, 2015; Van Loon et al., 2016).
Altogether these influences cause a strong variability of climatic and hydrological drought hazard in tropical
space (Nauditt et al., 2019b).

Daily time step data, needed to effectively evaluate drought hazard in tropical catchments, are rarely
available. Sheffield et al. (2018) highlight the potential of satellite remote sensing and reanalysis data
products to improve water resources management in regions with sparse in-situ monitoring networks. Open
access high resolution remote sensing data products are continuously increasing in quantity (AghaKouchak
et al., 2015; Mariano et al., 2018). In this context, a variety of gridded datasets are available, including daily
precipitation (Funk et al., 2015; Baez-Villanueva et al., 2018&2020), surface water (Beck et al., 2016),
groundwater (Thomas et al., 2014), reservoirs (AghaKouchak et al., 2018), soil moisture (Samaniego et al.,
2018; Tijdeman and Menzel, 2020), and vegetation (Pinzón and Tucker, 2014; Nguyen et al., 2019).
Especially vegetation condition indices like fAPAR, NDVI, EVI and VCI play an increasing role for drought
monitoring and research in data scarce regions. They can provide spatially distributed information on soil
and vegetation moisture anomalies on the ground (Heydari et al., 2018; Recuero et al., 2019) that is not
dependent on sparsely monitored hydro-climatic data.



Exposure and vulnerability information are also sparse, especially in rural tropical regions. Vulnerability evaluation should be ideally based on historical drought impact data (Stahl et al., 2016; Bachmair et al., 2016; Blauhut et al., 2016), but these are usually not systematically monitored and recorded; rare examples being the US Drought Impact Reporter (droughtreporter.unl.edu) , the European Drought Impact Database (Stahl et al., 2016) or observer-based systems such as the Czech INTERSUCHO (www.intersucho.cz). Alternatively, vulnerability data is often replaced by exposure related information (Carrão et al., 2016; Naumann et al., 2018; Vogt et al., 2018; Naumann et al., 2019), that is available as gridded socioeconomic data sets showing the spatial distribution of population-, livestock- and crop densities as well as socio-economic, demographic and infrastructural characteristics. Such remote sensing and gridded data-based drought risk assessment approaches have often been carried out at global or regional scale (Carrão et al., 2016; Hagenlocher et al., 2019), but have rarely been applied to local and catchment scale drought risk. This study evaluates the performance of gridded datasets related to hydro-climatic and socio-economic information to derive relevant drought risk information for catchments of different sizes (between 5 450 and 49 382 km²) and differing tropical climates.

In line with the above, **the overall aim** of this study is to identify and characterize drought risk hotspots in rural and data scarce tropical regions as a basis for drought management.

**Specific objectives:**
- to identify the spatially distributed cumulative duration of hydrological and meteorological drought hazard

- to understand spatially varying and sector related drought vulnerability

- to visualize spatial distribution of drought risk in four tropical catchments that vary in size, topography, climate and water infrastructure development

- to attribute the relative spatial contribution of hazard and vulnerability related factors to drought risk



## 2 Data and Methods

### 2.1 Study regions

We selected four rural catchments in tropical regions that were affected by severe drought disasters during the last decade. Figure 1 gives an overview on the characteristics of the four *study regions*, each differing

in size, topography seasonality and level of human intervention.

**Figure 1.** *Study regions: river network, discharge stations, major land uses and urban areas of the (A) Muriaé, subbasin of the Paraíba do Sul in South Eastern Brazil, (B) the Tempisque basin in North Costa Rica, (C) the Upper Magdalena basin in Colombia and (D) the Srepok basin in the Lower Mekong shared by Cambodia and Vietnam.*




**Table 1: Catchment characteristics of the four study regions**

| | The Muriaé catchment (Paraíba do Sul River Basin), South Eastern Brazil | The Tempisque-Bebedero catchment, Costa Rica | The Upper Magdalena basin, Colombia | The Srepok basin, Lower Mekong (Cambodia and Vietnam) |
|---|---|---|---|---|
| **Size:** | 7 220 km² | 5 455 km² | 49 382 km² | 30 900 km² |
| **Elevation:** | 10 to 2 000 m.a.sl. | 0 to 1 916 m.a.sl. | 222 to 3 685 m.a.sl. | 66 to 2 283 m.a.sl. |
| **Precipitation:** | 1 000–2 000 mm. | 1 000-3 000 mm. | 2 500-3 000 mm | 1 569–2 800 mm |
| **Mean annual discharge:** | 118 m³/s. | Tempisque 27 m³/s; Bebedero 10 m³/s. | 1 330 m³/s. | 634.2 m³/s. (Constable, 2015) |
| **Total population:** | ca. 100 000 inhabitants | ca 382 900 inhabitants. | ca. 1.5 inhabitants | ca. 2.9 million inhabitants |
| **Climate:** (Peel et al., 2007) | Tropical savanna climate (Aw), dry-winter humid subtropical climate (Cwa) and Dry-winter subtropical highland climate (Cwb). | Tropical savanna climate (Aw). | Tropical rainforest climate (Af), tropical monsoon climate (Am), oceanic climate (Cfb), and tropical savanna climate (Aw). | Tropical savanna climate (Aw) and tropical monsoon climate (Am). |
| **Major land uses:** (Arino et al., 2012) | 68.3 % pasture, 24.1 % forest, 7 % agriculture and 0.6 % urban. | 17 % pasture, 5 % forest, 76 % agriculture and 2 % urban. | 38 % agricultural area, 51 % forest, 9 % pasture and 1 % urban areas. | 40 % pasture, 34.5 % forest, 24.9 % agriculture and 1 % urban. |

## 2.2   Data

**Discharge data set: Hydrostreamer**

Available discharge observations data in the study regions (Figure 1) do not allow to display the spatial variability in hydrological behaviour. We applied a recently developed downscaling tool, Hydrostreamer (Kallio, 2020) to the spatially coarse global discharge data product from the ISIMIP 2a (Gosling et al., 2017) experiment. Downscaling is carried out by areal interpolation, where the source runoff data are distributed to intersecting higher-resolution catchments, routed downstream, and optimized against observed streamflow

(see detailed description in Kallio et al., 2019&2020. For the Muriaé catchment, SWAT2012 modelling (Neitsch et al., 2011) provided 93 simulated discharges used for optimization (Nauditt et al., 2019b).

We requested available daily discharge observations for the optimization of the Hydrostreamer results. For the Muriaé, five daily discharge time series were obtained by the National Water Agency of Brazil (ANA,


2019; Nauditt et al., 2019a; Nauditt et al., 2019b). For the Tempisque, data from two discharge stations were acquired by the Hydrological Department of the Electricity Institute of Costa Rica ICE (2019) 1980-2003 and for the period 2003-2018 by the Institute for Aqueducts and Sanitation (AYA, 2019). For the Upper Magdalena, we obtained daily time series from 46 discharge stations from IDEAM (2019) and for the Srepok, daily data from three discharge stations (Kallio et al., 2019; Kallio, 2020; MRC, 2018) were acquired.

**Precipitation**

Station data that were used to validate the satellite based precipitation estimates for the Magdalena and the Muriaé are described in Baez-Villanueva et al. (2018), for the Srepok in Dandridge et al. (2019) and for the Tempisque in Venegas-Cordero et al. (2020). We selected the gridded 0.05° x 0.05° resolution daily precipitation dataset CHIRPS v2.0 (Funk et al., 2015) for the period 1980-2018 after evaluating different satellite based precipitation data sets in a point to pixel analysis (Baez-Villanueva et al., 2018; Dandridge et al, 2019) and in hydrological modelling (Nauditt et al., 2019b).

CHIRPS v2.0 showed good goodness-of-fit (GOF) performance in point to pixel evaluation and HBV rainfall runoff modelling (Baez-Villanueva et al., 2018; Nauditt et al., 2019b; Venegas-Cordero et al., 2020). Additionally, CHIRPS v2.0 covers the longest time period (1981 to date) and has a higher spatial resolution compared to other available satellite based precipitation estimates products.

**Vegetation Condition**

To understand the spatial variation of drought related vegetation condition, we used MODIS MOD13Q13 (2000-2017) 16-day composite NDVI imagery at 250 m resolution. We identified the driest month in record using the Standardized Precipitation Index (SPI) (McKee et al., 1993) applied to the satellite based precipitation raster data set CHIRPS v2.0 (1980-2017) (Nauditt et al., 2019b).


**Vulnerability**

We evaluated gridded data sets in terms of their suitability to represent vulnerability and selected the following data sets available for all our study regions:

**Table 2: Vulnerability (Exposure) data sets**

| Data set | Vulnerability Indicator | Source |
|---|---|---|
| Gridded Livestock of the World (GLW) | livestock density | (Robinson et al., 2014) |
| Global Agricultural Lands 2000 | cropland density | (Ramankutty et al., 2008) |
| GHS Population Grid 2015 | population density | (CIESIN, 1997-2020) |
| Major roads 2013 | proximity to infrastructure | (CIESIN, 1997-2020) |
| Global GDP PPP/HDI 2011 | GDP | (Kummu et al., 2018) |


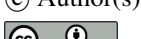


## 2.3    Drought Risk Assessment

**Figure 2** illustrates the overall methodology applied in this study. We evaluate drought risk as a combination of hazard and vulnerability:

$$dr_i = \frac{dh_i + dv_i}{2} \tag{1}$$


Where $dr$ represents drought risk, $dh$ drought hazard, $dv$ drought vulnerability and $i$ grid cell.

Hazard ($dh_i$) is defined by drought in meteorological, hydrological and vegetation condition indices. Vulnerability ($dv_i$) is defined as the potential of a drought to cause damages in selected socioeconomic sectors using typical exposure information.

We used two groups of variables (hydro-climatic and socio-economic) and calculated index values for each grid cell (*i*) for each variable. All layers were resampled to a spatial resolution of 30 m and equally weighted to obtain maps for each index, hazard, vulnerability and risk. Drought risk maps were then produced by equally weighting the hazard and vulnerability layers. More details about the methodological process are given in sections *2.3.1* and *2.3.2*.





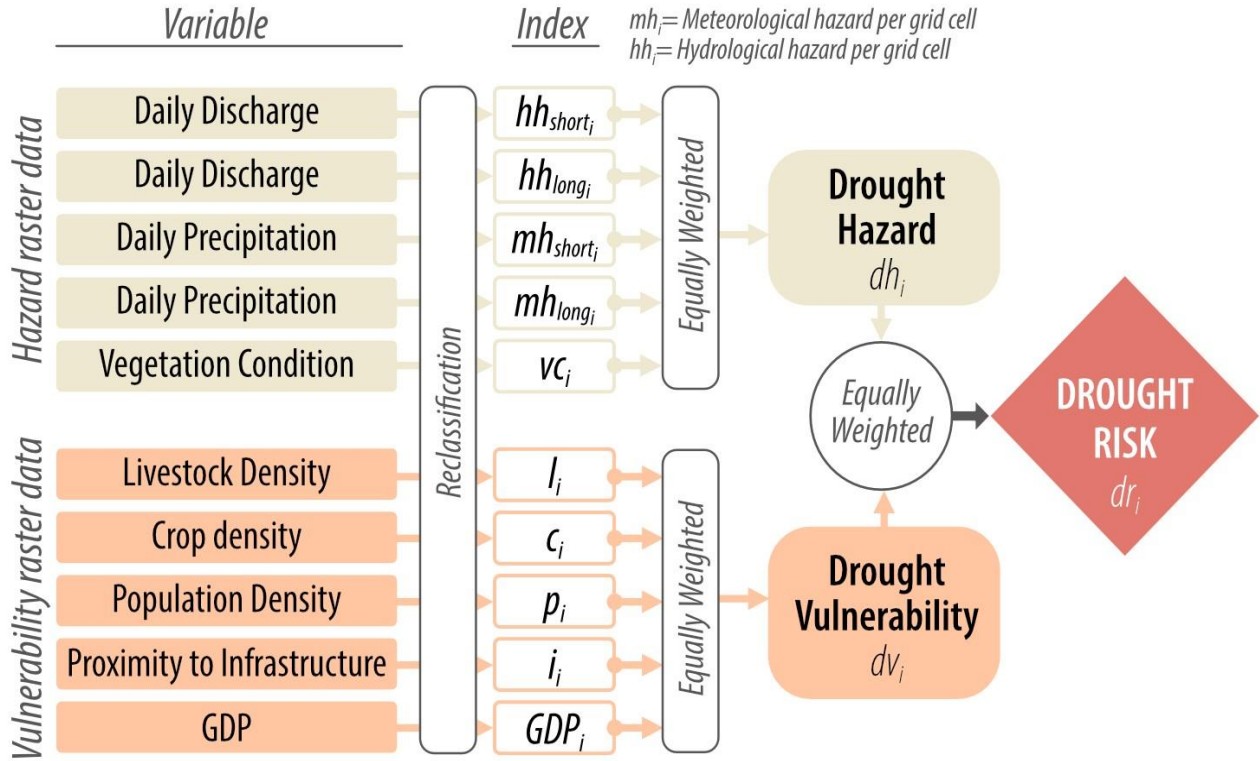

$mh_i$= Meteorological hazard per grid cell
$hh_i$= Hydrological hazard per grid cell

**Figure 2.** *Overall methodology (for variable and index descriptions see sections 2.3.1 and 2.3.2)*

### 2.3.1 Drought Hazard

To obtain daily scale **hydrological drought** signals, we applied the widely used threshold method (e.g. Tallaksen, 2000) using a daily varying $Q_{95}$ threshold (Fleig et al., 2006; WMO, 2008). We selected the period 1981-2018 that corresponds to the record length of CHIRPS v2.0 data. We defined more or equal than 12 days below a daily varying $Q_{95}$ threshold as a **long** hydrological tropical drought event ($hh_{long}$) and 5-11 days below that threshold as a **short** hydrological tropical drought event ($hh_{short}$). We used pooling to remove single days when streamflow went above the threshold by less than 20 %. Resulting short and long hydrological drought indices ($hh_i$) were derived as the cumulative drought duration of events for each grid cell. So ($hh_{short_i}$) is the sum of all short-duration (5-11 days) events and ($hh_{long_i}$) is the sum of long-duration (>=12 days) events. The cumulative duration of detected events was classified into five severity categories (*Sc*). More than 75 short drought events during the period of 37 years were classified as the most severe




short drought hazard and more than 50 events with more or equal than 12 days below $Q_{95}$ were considered the most severe long drought hazard (*Sc* 5) (Table 3).

The **meteorological drought** index $(mh_i)$ evaluates the cumulative drought duration of precipitation drought

events. To represent long and short meteorological drought events in tropical regions, we defined two classes of drought intensity for precipitation deficits: >= 20 days with rainfall below 0.3 mm as a long meteorological drought: $(mh_{long})$ and 5-19 days as a short meteorological drought: $(mh_{short})$ with rainfall below 0.3 mm. The number of detected events were classified into 5 severity categories (Table 3)

The **vegetation condition** related drought hazard *vc* is represented by the vegetation condition in the driest

month in record. We identified the driest month in record using the SPI. To understand the spatial variation of vegetation condition, we used the Vegetation Condition Index (VCI) (Dutta et al., 2016; Kogan, 1995; Quiring and Ganesh, 2010)  applied to NDVI imagery. The vegetation related drought index (*vc$_i$*) was established by inversely rating VCI values for each pixel. In contrast to the hydrological and meteorological indices, *vc$_i$* has a negative correlation with drought severity. Values between 50 % and 100 % indicate

moisture rich vegetation conditions, values between 50 % and 35 % short drought conditions and values below 35 % long drought conditions (Kogan, 1995). The detailed methodology is described in Nauditt et al., 2019b. VCI percentage values were classified into five severity categories (Table 3).

The overall drought hazard (*dh*) for each grid cell (*i*) is calculated by the equally weighted severity class (*Sc*) values (Table 3) of each hazard index as:


$$dh_i = \frac{S_c\left(hh_{short_i}\right) + S_c\left(hh_{long_i}\right) + S_c\left(mh_{Short_i}\right) + S_c\left(mh_{long_i}\right) + S_c(vc_i)}{5} \qquad (2)$$

Where *dh* is the drought hazard, *$_i$* the location (grid cell) and *Sc* the severity class. $hh_{short_i}$ represents the cumulative duration of short hydrological drought events based on number of events, $hh_{long_i}$ the cumulative

duration of long hydrological drought events, $mh_{short_i}$ the cumulative duration of short meteorological drought events, $mh_{long_i}$ the cumulative duration of long hydrological drought events and $vc_i$ the vegetation condition related hazard (Table 3).





**Table 3** *Hazard indices, their severity classification and allocation to five severity classes (Sc):*

| Drought Hazard Index | $hh_{short_i}$ | $hh_{long_i}$ | $mh_{short_i}$ | $mh_{long_i}$ | $vc_i$ |
|---|---|---|---|---|---|
| | Number of drought events with 5-11 days below $Q_{95}$ daily variable threshold | Number of drought events with >=12 days below $Q_{95}$ daily variable threshold | Number of drought events with 5-19 days below 0.3 mm of precipitation | Number of drought events with >=20 days below 0.3 mm of precipitation | Vegetation Condition Index (VCI) value (%) for the driest month in records (SPI12) |
| Severity class (Sc) | Classification | | | | |
| 1 | 0-30 | 0-25 | 0-30 | 0-25 | > 50 |
| 2 | 30–45 | 25–32 | 30–45 | 25–32 | 40-50 |
| 3 | 45–60 | 32–40 | 45–60 | 32–40 | 30-40 |
| 4 | 60–75 | 40–50 | 60-75 | 40–50 | 20-30 |
| 5 | > 75 | > 50 | > 75 | > 50 | 0-20 |


### 2.3.2   Drought Vulnerability

We used open access gridded datasets for five socioeconomic exposure related variables to represent spatial drought vulnerability in the four study regions. All datasets were resampled using the nearest neighbor method to account for differences in grid cell resolution. Each data set was reclassified and given a rating

based on positive or negative correlation to vulnerability. The overall drought vulnerability *dv* for each grid cell *i* is calculated by the equally weighted severity class *Sc* values (Table 4) of each vulnerability index as:

$$dv_i = \frac{S_c(l_i) + S_c(c_i) + S_c(p_i) + S_c(i_i) + S_c(GDP_i)}{5} \qquad (3)$$

Where $dv_i$ *is* overall drought vulnerability per grid cell, *Sc* the severity class, $l_i$ the livestock density index, $c_i$

the crop density index, $p_i$ the population density index, $i_i$ the index for proximity to infrastructure and *GDP$_i$* the GDP index per grid cell. Table 4 gives an overview on the severity classification for each index.




**Table 4 Vulnerability indices, classification and severity classes**

| Vulnerability index | $l_i$ Livestock density (head per km²) | $c_i$ Cropland (% area) | $p_i$ Population Density (persons/grid cell) | $i_i$ Proximity to Infrastructure (m) | GDP$_i$ (Million USD PPP per km²) for reference year 2011 |
|---|---|---|---|---|---|
| **Severity class** | Classification | | | | |
| **1** | 0-15 | 0-0.1 | 0-50 | 0-100 | >20 |
| **2** | 15-30 | 0.1-0.2 | 50-100 | 100-250 | 5-20 |
| **3** | 30-40 | 0.2-0.3 | 100-250 | 250-500 | 2-5 |
| **4** | 40-50 | 0.3-0.4 | 250-500 | 500-1000 | 1-2 |
| **5** | >50 | >0.4 | >500 | > 1000 | 0-1 |
| **Correlation** | positive | positive | positive | positive | negative |

## 3   Results

### 3.1   Drought hazard ($dh_i$), vulnerability ($dv_i$) and risk ($dr_i$) in the four study regions

Figure 3 gives an overview on spatial coverage (% of grid cells) of drought hazard, vulnerability and risk in the four study regions. Due to the equal weighting of the individual hazard and vulnerability severity values, the percentages of basin area in the severity classes 1 and 5 are small. Therefore, *Sc 4* can be considered as most severe and *Sc 2* as least severe.

For the **Muriaé**, results suggest severe vulnerability (*dv*) in most of the area with 73.1 % in *Sc* 4 and 26.2 % in *Sc* 3. Hazard (*dh*) was found in *Sc* 3 with 57.4 % and *Sc* 2 with 24.4 % of the Muriaé basin. The **Tempisque** showed hazard (*dh*) with 25.7 % of the area in *Sc* 4, 0.7 % in *Sc* 5 and 56.5 % in *Sc* 3, while percentage of area exposed to drought vulnerability (*dv*) is largest in *Sc* 3 with 81.8 % and 14.2 % in *Sc* 2. **Upper Magdalena** shows vulnerability (*dv*) with 43.1 % in *Sc* 4 and 55 % in *Sc* 3. Results identify hazard (*dh*) with most values in *Sc* 2 with 52 % and 41.3 % in *Sc* 3.  Finally, for the **Srepok,** higher hazard (*dh*) was found compared to *dv*, with 38.5 % in *Sc* 4 and 54 % in *Sc* 3 of its basin area, while vulnerability is highest in *Sc* 3 with 87.4 % and 8.9 % in *Sc* 2 (Figure 3).





**Figure 3 Severity class (Sc) distribution per percentage of area for drought hazard (*dh*),
vulnerability (*dv*) and risk (*dr*) in the four study regions**





Figure 4 shows balloon plots indicating the percentage of the basin area covered by the severity classes 1-5 for each drought index. All study regions show a high vulnerability related to low GDP ($Sc$ 5) and a low vulnerability ($Sc$ 1) related to population density ($p_i$).

For the **Muriaé,** highest severity was found for livestock ($l_i$) (with 78 % in $Sc$ 5) and crop density ($c_i$) (33.2 % in $Sc$ 5 and 31.1 % in $Sc$ 4) as well as for proximity to infrastructure ($i_i$) (43.5 % in $Sc$ 5 and 25.7 % in $Sc$ 4). The remaining indices showed a nearly homogenous distribution across the severity classes.

For the **Tempisque**, results show highest values for short hydrological drought hazard ($\boldsymbol{hh_{short_i}}$) with 42 % of its area in $Sc$ 4 and 10,5 % in $Sc$ 5, for $\boldsymbol{mh_{short_i}}$ with 33 % of its area in $Sc$ 4 and 26 % in $Sc$ 5 and for

$\boldsymbol{hh_{long_i}}$ with 23 % of its area in $Sc$ 5. Lower values were shown for the vulnerability indices $l_i$ with 25.1 % of area in $Sc$ 4 and 19.1 % in $Sc$ 5 and $c_i$ (44.2 % of area in $Sc$ 4 and 17 % in $Sc$ 5) (Figure 4).



**Figure 4** *Index values distributed to percentage of catchment area in the four study regions*





For the **Upper Magdalena,** highest severity was found for vulnerability indices crop density ($c_i$) (55.6 % in
*Sc 5* and 18.1 in *Sc 4)*, livestock density ($l_i$) (21.4% in *Sc* 5 and 16.7 *%* in *Sc 4)* and proximity to infrastructure
(36.2 % in *Sc* 5 and 24.8 *% in Sc* 4). Lower values for hazard were found for $mh_{short_i}$ with 37 % in *Sc* 3 and
35.8 % in *Sc* 4, for $mh_{long_i}$ (11. 6 % in *Sc* 5 and *39.2 %* in *Sc* 4*)* and very low values for $hh_{long_i}$ (73.9 % in
*Sc* 1 and *15.4 %* in *Sc* 2*) and* $hh_{short_i}$ (71.8 % in *Sc* 1 and *17.2 %* in *Sc* 2*).*

For the **Srepok**, results show highest values for vegetation condition related condition hazard $vc_i$ (46.4 % in
*Sc* 5 and 15.9 % in *Sc* 4), followed by $mh_{long_i}$ (*50.1 %* in *Sc* 4 and 5.8 % in *Sc* 4*).* The other hazard indices
resulted in values distributed across the *Scs.* Highest vulnerability was found for proximity to infrastructure
(*i_l*) with 51.5 % in *Sc* 5 and 20.6 % in *Sc* 4. The remaining vulnerability indices showed low values as for
livestock density ($l_l$) 88.2 % in *Sc* 1 and for crop density ($c_i$) 75.4 % in *Sc* 3.

### 3.2    Spatial distribution of drought hazard (*dh_i*) in the four study regions

Figure 5 illustrates the weighted drought hazard based on the meteorological, hydrological and vegetation
condition-based indices for each grid cell in the four study regions. Red corresponds to *Sc* 5 and green to *Sc*
1. For the **Muriaé**, drought hazard was found to be highest in the Southwestern downstream part due to the
larger share of *Sc* values for $hh_{short_i},$ $mh_{long\ i}$ and $vc_i$ (see individual maps in the supplementary material).

For the **Tempisque** basin, drought hazard was found to be highest in the downstream part in the North over
the river estuary and for the Eastern part along the main Tempisque River stretches and the Bebedero
tributary upstream (Figure 5). Duration of periods without rainfall and resulting vegetation moisture loss were
stronger in the Eastern part of the basin.

For the Southwestern upstream area of the **Upper Magdalena** basin in Colombia, the results show strongest
hazard values in the Southeastern upstream part where most meteorological drought periods and also
vegetation related anomalies were detected and along the main Magdalena river due to hydrological
droughts. For the **Srepok** basin, the strongest hazard values were observed in the Southeastern upstream
region over the Vietnamese highlands and the Cambodian North and Northwestern downstream region due
to high vegetation condition hazard *v_i* and hh *long* (Figure 5). Detailed results for each layer are given in the
supplementary materials.








**Figure 5** *Spatially distributed drought hazard (dh) severity classes found in the four study regions*



### 3.3    Spatial distribution of drought vulnerability

Figure 6 shows the spatial distribution of drought vulnerability in the four study regions. For the **Muriaé**, the data evaluation suggests a strong vulnerability all over the region due to high cropland and livestock density,

low GDP of the rural population and long distances to infrastructure - except along the major roads where the indicators for crop density and proximity to infrastructure show low values. Results for the **Tempisque** show a high vulnerability almost all over the place as cropland and livestock pasture is strongly developed. Only the protected areas in the forested Northwestern and Central regions show lower vulnerability values. Similarly, for the **Upper Magdalena** drought vulnerability was found almost everywhere, despite the

Southwest, where protected areas are located. In the **Srepok,** less vulnerability seveirity was found, with main regions located in the Vietnamese Southeastern upstream part, the Northeaster Cam bodian part and the Northwestern downstream area (Figure 6). Detailed results for each layer are presented in the supplementary materials.

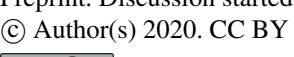


Figure 6 *Spatial distribution of drought vulnerability (dv) severity classes in the four study regions*






## 3.4 Drought risk

Figure 7 illustrates the spatial distribution of drought risk (*dr*) in the four study regions based on equally weighted hazard and vulnerability severity values.

**Figure 7** *Spatial distribution of drought risk (**dr**) severity classes in the four study regions*



In all study regions, results show a considerable share of area with drought risk hotspots (with colours from orange to red). While in the **Muriaé,** highest risk was observed in the meteorologically drier downstream part; the **Tempisque** showed the strongest risk in the central-eastern and estuary region, where both hazard and vulnerability values were found to be severe (see individual maps in supplementary materials). For the **Upper Magdalena**, severe drought risk was found in the central Magdalena valley from up- to downstream (Northeast) and in the meteorologically dry Southeastern upstream part.

The **Srepok s**howed severe risk in the Southeastern Vietnamese upstream region and in the Northern and Eastern central region of Cambodia.

## 4    Discussion

### 4.1    Hazard ($dh$), vulnerability ($dv$) and drought risk ($dr$) in the four study regions - plausibility of identified drought risk hotspots

For the **Muriaé**, plausible results were obtained for the location of drought risk hotspots as well as for the spatial distribution of severe drought hazard and vulnerability. Risk hotspots were found in the downstream area where most economic activities take place and precipitation rates are lower compared to those at higher elevations (CEIVAP, 2015; Nauditt et al., 2019b), along with a higher hydrological hazard also due to fractured geological and alluvial characteristics. Vulnerability is high all over the basin due to intensive livestock grazing and agricultural production as well as low GDP and large distances to road infrastructure. These spatial characteristics for drought hazard, vulnerability and risk were confirmed by collaborating stakeholders of the river basin committee CEIVAP (Comitê de Integração da Bacia Hidrográfica do Rio Paraíba do Sul) and the executive river basin agency AGEVAP (Agência da Bacia do Rio Paraíba do Sul). Additionally, results coincide with field research, modelling and data analysis related to spatial variability of drought occurrence and impacts of involved and affiliated scientists (Nauditt et al., 2019a).

Also the locations of drought risk hotspots in the **Tempisque** were well defined by the analysis. Hotspots were found in the Northwestern downstream part in the river estuary and in the Southeastern part along the main river stretches and the Bebedero tributary upstream. These regions show high values for both, hazard and vulnerability. Hazard is high due to human abstractions along the streams and irrigated areas and due to longer periods without rainfall and resulting vegetation moisture loss in the Eastern part of the basin. Risk hotspots (Figure 7) are found where this hazard is combined with highest vulnerability due to highest crop and livestock density.  Lower $dr$ values were found in the Northeast, where vulnerability is low in larger



National Parks. The results are in agreement with information of the Water Agency of Costa Rica DA (Dirección de Agua de Costa Rica) and the team of the PIAAG Program (Programa Integral de

Abastecimiento de Agua para Guanacaste – Pacífico Norte) that aims at securing the water supply to the Tempisque Bebedero region (Dirección de Agua, 2018). These findings are supported by comprehensive (field) research in the drought prone study region on topics related to meteorological and hydrological droughts, water scarcity and vegetation susceptibility to droughts (Muñoz Jiménez et al., 2019).

For the **Upper Magdalena Basin** in Colombia, drought risk hotspots were found in the Northeastern

downstream part, all over the Magdalena valley and in the Southwestern upstream area (Figure 7). While agricultural activities and related water abstractions increase both vulnerability and hydrological hazard in the central valley, the risk hotspots in the upstream can be explained by strong meteorological hazard in the Southwest, which is drier and where most meteorological drought periods and vegetation related anomalies were detected. Hydrological droughts in the main stream are most probably aggravated by hydropower

operation and abstractions for rice irrigation (Vega-Viviescas and Rodriguez, 2019). However, in total terms, the Upper Magdalena shows low total hazard per percentage of area with most values in Severity Class 2 (52 %). This can be explained mainly by the few grid cells affected by strong hydrological hazard along the major streams of the Magdalena. Vulnerability in the Magdalena was found to be more relevant compared to hazard with 43.1 % in *Sc* 4 and 55 % in *Sc* 3. This coincides with the spatial distribution of main areas of

crop, livestock and population density in the Magdalena valleys and lower lying areas, as confirmed by scientists at Universidad Nacional de Colombia with vast research experience in the study region as well as by stakeholders as the Colombian Agency for Hydrology and Meteorology IDEAM and the Magdalena Basin Agency Cormagdalena.

Drought risk hotspots were also well detected in the **Srepok** basin, with main locations observed in the Southeastern upstream region over the Vietnamese highlands and the Cambodian North and Northwestern downstream region. High risk values depended mainly on high vegetation condition hazard ($v_i$) and hydrological hazard $hh_{short}$ and $hh_{long}$ that is dominating in the upstream area where reservoirs are located and along the downstream rivers, from where abstractions are used for irrigated rice in Cambodia (Bui Du,

2018; Constable, 2015). Vulnerability is lower compared to the other study regions, as approximately 50 % of the basin is covered with forest and due to the absence of livestock. Higher values for *dv* are only found in the Northwestern region, where agricultural land in Vietnam is cultivated with cash crops, mainly coffee (DaLat), rubber, cashew, black pepper and fruit trees for domestic and export markets (CCAFS-SEA). Results were confirmed by scientist of Aalto University with years of research experience in the region, as





well as by collaborating institutions (Ministry of Environment of Vietnam, MONRE and Water Management
Institute NAWAPI).

### 4.2    Drought hazard assessment

In the **Muriaé**, severe hydrological hazard $hh_{short_i}$ was found in the Northeastern agricultural upstream area,
along the streams and at the river basin outlet with 20 % in *Sc* 5 and another 20% in *Sc* 4. During the long-
term drought in 2014-2015, the river stretch near the catchment outlet station fell dry (Nauditt et al., 2019b;
Ribbe et al., 2018), with impacts on aquatic and riparian ecosystems and water users. This event was
aggravated due to the fractured geological and alluvial characteristics of the downstream river bed (Nauditt
et al., 2019b). Cumulative duration of $hh_{long_i}$ with 12 or more days however, was only present in 14.1 % of
area in *Sc* 4 and 1.8 % in *Sc* 5 at the aforementioned basin outlet. The Northeastern agricultural upstream
area in Minas Gerais State is prone to $hh_{short_i}$, most probably due to smaller catchment areas and fast
response to rainfall deficits.

Hydrostreamer provided excellent spatially distributed discharge simulations for the Muriaé catchment, as
validated by station data and SWAT2012 modelling results for 93 stations; very valuable for drought and
water resources management and planning. In the Southwestern downstream part severe meteorlogical
hazard (*Sc 5)* was found for $mh_{short_i}$, $mh_{long_i}$ for most grid cells. Low values were found for the upstream

region (*Sc*1 for most grid cells). $vc_i$ is following this spatial pattern with low hazard in the upstream and
hazardous vegetation condition in the downstream area. This shows that not only the magnitude of rainfall
in the mountainous upstream region (2000 m maximum elevation) was greater (Künne et al., 2018) compared
to the drier downstream catchment, but also many less events with consecutive days without rainfall
($mh_{short_i}$ and $mh_{long_i}$) occurred in the period between 1981 and 2018 (see individual index maps in the

supplementary materials). Both regions are extremely vulnerable to meteorological droughts: while in the
upstream part in Minas Gerais rainfed horticulture is dominating, downstream, in Italva, Rio de Janeiro State,
livestock and milk production is the main economic activity (Fischer et al., 2018).

Hydrological drought hazard was found to be most severe along the upper streams of the **Tempisque** from
which irrigation water is abstracted. This shows that our approach to assess hydrological hazard with a daily
varying threshold is also suitable for anthropogenically intervened catchments, where human abstractions
are leading to discharge anomalies – most probably increasing as a response to a meteorological drought.
23.1 % of the basin area experienced more than 40 long drought events that lasted longer than 12 days ($Q_{95}$





of daily varying discharge and below) and 10.5 % of its area was affected by 60 moderate drought events in the time period between 1918 and 2018. This shows the extreme low flows (down to 2.6 m³/s at Guardia station) the drought prone region is facing. A drought threshold of Q95 can therefore be considered as too low for a stream with a mean annual discharge of 27 m³/s in the case of the Guardia Station.

Strong spatial variation in meteorological drought hazard was found for $mh_{short_i}$ and $mh_{long_i}$, with high values

(*Sc* 5) covering the Eastern part of the basin (Bebedero subcatchment) and values of *Sc* 1-2 dominating the Western region (Tempisque). $vc_i$, in contrast, is homogenously distributed all over the basin, probably as the NDVI image was taken during a dry anomaly (SPI 12) of a dry season. Although both total annual and dry season rainfall (December-May) accounts for similar monthly precipitation values in both regions (Bocanegra, 2017), results of our study show a much larger number of both short and long meteorological

drought events in the Eastern Bebedero subcatchment compared to the Tempisque (see individual index maps in the supplementary materials).

In the **Upper Magdalena** only few grid-cell values for long and short hydrological hazard were identified along the major streams of the Magdalena. Hydrological droughts in the main streams are most probably aggravated by hydropower operation and abstractions for rice irrigation (Vega-Viviescas and Rodriguez,

2019). However, in total terms, the Upper Magdalena shows low hydrological hazard per percentage of area with most values in Severity Class 2 (52%). This might be due to data uncertainties in the hydrostreamer dataset and the underlying observed discharge data (Rodríguez et al., 2020). Hydrostreamer yielded in poorer performance compared to the other three study regions.

The Southwestern upstream region showed strongest meteorlogical hazard (*Sc* 5) for $mh_{short_i}$ and $mh_{long_i}$

followed by the Northeastern downstream part and similar spatial patterns for *vc$_i$*. The Southwestern upstream region is exposed to a more marked tropical seasonality with two wet periods (April and May and October and November) and two long dry periods (June to October and November to April) (Rodríguez et al., 2020) while the lower part of the Magdalena receives more precipitation and is not exposed to such a marked seasonality.


For the **Srepok** basin, hydrological drought hazard for both *hh$_{long}$* and *hh$_{short}$* was found to be most severe in the Vietnamese Southern upstream region in the Vietnamese highlands due to discharge alterations through hydropower operation, as well as in the Cambodian North-Central and Northwestern downstream region due to abstractions from agricultural activities.





Good results for Hydrostreamer downscaling results were obtained for the Srepok, being a study region of
        Kallio et al. (2019 & 2020), providing a valuable discharge data set for water resources modelling,
        management and planning in the transboundary basin and the Mekong region.

        Meteorological hazard was strong with 55.9 % in *Scs* 4 and 5 for $mh_{long\ i}$, with all grid cells located in the
        Vietnamese Southeastern upstream part. $mh_{short_i}$ was only detected in Scs 1-3, indicating that periods

without rainfall of shorter duration during the wet period were less frequent. $vc_i$, in contrast, is homogenously
        distributed all over the basin.

### 4.3     Vulnerability assessment

        We applied open access gridded data sets to evaluate their suitability to provide drought vulnerability or
        exposure information for all of the four study regions. For all study regions, our indicator population density

(*$p_i$*) showed few grid cells with severity values higher then *Sc 1* (classified as less than 50 persons per 1 km²
        grid cell). This suggests that the classification we chose (Table 4), assuming that >50 persons would
        represent small settlements and agricultural communities in rural regions, might not be adequate. The
        number of persons per km² classified as vulnerable could be lower to also detect remote farmers. In contrast,
        low GDP in rural areas showed strongest severity (*Sc 5* = < 1 million USD PPP per km² for the reference

year 2011) for almost all grid cells in all study regions, outweighing the low *$p_i$* values. We used this
        classification assuming a low GDP for rural agricultural regions; however, our results suggest that higher
        classification values for *Sc 1* and *Sc 2* in order to display differences in GDP. In most risk studies, several
        exposure and vulnerability indicators are aggregated, regionally masked (Naumann et al., 2014, Carrão et
        al., 2016; Hagenlocher et al., 2019) to show overall vulnerability. Reference values for indicator classification

– at least to our knowledge -- are not available in literature. Generally speaking, a more detailed evaluation
        of gridded socioeconomic data in terms of risk indicator classification, e.g. for tropical agricultural regions
        resulting in reference values would be a valuable contribution to future comparative risk studies.

        To evaluate drought exposure and vulnerability of agricultural activities, we tested the data set "Global
        Agricultural Lands in the Year 2000". The resulting crop density evaluation, similarly as population density,

is mostly distinguishing between grid cells with agriculture and no agricultural use, therefore a low
        classification values were used (Table 4) resulting in a good representation of agricultural exposure in the
        four study regions, as confirmed by affiliated scientists and stakeholders. More detailed local information on
        crop types, eg. distinguishing between perennial crash crops and annuals, irrigated or non-irrigated
        agriculture, could further detail such site-specific exposure information. Similarly for Livestock density related

vulnerability, we used a low number of animals grazing per grid cell to determine the low *Scs* (Table 4).





Proximity to Infrastructure served as a good proxy for the stage of development of a location. Although drought vulnerability and exposure largely depends on storage infrastructure or irrigation systems, there are no available data sets for the regions addressed in this study. FAO AQUASTAT, for example, provides such data for Africa but not for Latin America and South East Asia. Despite these shortcomings, our overall drought

vulnerability index $dv_i$ showed good results for aggregated vulnerability. The Muriaé had most grid cells in high severity classes (73.1 % in *Sc 4*) due to its prevailing sectors milk production and agriculture as well as its sparse road infrastructure. For the Tempisque, we found fewer – but well distinguished and located -- grid cells in *Sc 4* (81.8 % in *Sc 3*); because of the existence of large National Parks in the Northeastern part, less cattle grazing and its well-developed road network. The Upper Magdalena, (with 43.1 % in *Sc4*) showed high

$dv_i$ values in the downstream part mainly due to crop density and infrastructure. Only for the Srepok, less $dv_i$ was found. This can be attributed to little livestock and crop density and larger forested areas compared to the other study regions. $dv_i$ was mainly found for the cultivated areas along the streams as in Vietnam (upstream Southeast), irrigated rice areas in the Cambodian Northeast, and in the downstream Northwest.

## 5    Conclusion

Droughts are causing severe damages to water abundant tropical countries worldwide. The implementation of drought adaptation measures at the local scale need to be based on reliable information about spatially distributed drought risk -- that is rarely available in data scarce tropical catchments. We propose a methodology for evaluating and mapping the distribution of drought risk for rural tropical regions, based on the combination of independent indicators of daily data based drought hazard and drought vulnerability.

We evaluated freely available gridded datasets regarding their suitability to assess drought hazard, vulnerability and risk in four differing rural tropical study regions, the Muriaé river basin in South East Brazil, the Tempisque basin in Costa Rica, the upper catchment of the Magdalena river Basin, Colombia and the Srepok basin in Cambodia/Vietnam. We used daily scale meteorological and hydrological gridded data products and indices to evaluate tropical drought hazard, next to vegetation response to long term droughts,

as well as vulnerability data related to the major sectors population, agriculture, livestock, infrastructure and GDP that were available for Latin America and South East Asia. Results showed plausible spatial distribution of hazard, vulnerability and risk in the four study regions, as confirmed by local stakeholders, field surveys as well as through research of the authors. The following outcomes can be highlighted:

- The hydrological drought index $hh_i$, based on daily time series and a daily varying threshold $Q_{95}$, was
able to detect hydrological drought hazard in both, pristine and regulated streams, representing both climate and human induced hydrological drought.





- The meteorological drought index $mh_i$, based on daily precipitation data and periods of zero rainfall turned out to be suitable for tropical regions as shown by local impacts especially on livestock and rainfed agricultural production;

- The subindices $mh_{short_i}$ and $mh_{long_i}$ and $hh_{short_i}$ and $hh_{long_i}$ give insights in the historical frequency of long and short drought events, independent of general seasonal patterns.

- In combination with the above-described findings, the vegetation anomaly response (NDVI/VCI) to long term drought periods (SPI 12) reveals further vegetation, soil and groundwater related hazard, relevant for eg. forest fire related hazard.

- In light of the data scarcity in many tropical regions, the vulnerability related data sets and indicators related to crop and livestock density as well proximity to infrastructure have an adequate spatial resolution to provide vulnerability information at the local scale.

- The individual hazard and vulnerability indicator results give insights on how the classifications can be further adapted to individual study regions, depending on climate, topography, seasonality and human

influence.

- Drought hazard, vulnerability and risk maps and individual indicator maps provide decision support when selecting and designing drought adaptation measures to avoid future drought impacts.

These findings were discussed with representatives of local universities and public institutions working in the study regions, by looking at each indicator and combined hazard, vulnerability and risk. We recommend the

replication of the approach in other tropical regions by using the developed R scripts. A further step will be to make the Tropical Drought Risk Assessment R-Package available on CRAN.

Potential modifications to adapt or further develop our approaches are described below. These depend on the location of the study region, basin size, data availability, hydro-climatic seasonality and major socioeconomic uses. Individual indices can be adapted by changing the severity classification (e.g. by

increasing or reducing the duration of short and long droughts) or by introducing data sets that are not (yet) available for the regions addressed in this study. Furthermore, locally defined ecological flow and water demand thresholds could feed into a re-definition of local hydrological drought hazard indices. In addition, drought risk scenarios can be developed, helpful to detect potential changes in future drought risk. These could be based on hydro-climatic projections, indicating longer drought periods, or a changing vulnerability

based on socioeconomic projections.

However, in its current composition and design, our approach delivers a plausible representation of spatially distributed drought hazard, vulnerability and risk hotspots in data scarce and rural tropical regions. It offers a holistic, science based and novel solution to generate local drought risk knowledge that can feed into future



drought related research. The outcomes produced are a valuable source of information for regional planners
and water managers that take decisions on infrastructural and drought adaptation measures.

## 6    Code availability

R package is under preparation to be published on GitHub and possibly in Cran, in the meanwhile please
contact the corresponding author.

## 7    Supplement link

Hazard and vulnerability indicator maps

## 8    Author contributions

AN conceived and designed the study and overall methodology, performed the analyses and prepared the
manuscript. KS contributed to the study design, methodology, manuscript writing and structure. ER, CR and
RF supported field work, local validation of results and stakeholder communication in terms of research and
information demand to support drought management and validation. KM simulated and evaluated high
resolution discharge data (hydrostreamer). HH, OB and JT supported methodological setup, R Coding,
Package for GitHub and Cran and development of illustrations (R Graph Gallery), LR supported and
accompanied the research by allocating time and financial resources for personnel, travels and field work.
All authors actively took part in the writing and editing process.

## 9    Competing interests

The authors declare that they have no conflict of interest.

## 10    Acknowledgements

Field work scholarships, stakeholder workshops and travelling costs were supported by the CNRD Network
Project (www.cnrd.info) and the Tropiseca project (https://www.researchgate.net/project/TROPISECA-Multi-
lateral-University-Cooperation-on-the-Management-of-Droughts-in-Tropical-Catchments) funded by the
German Federal Ministry of International Cooperation (BMZ)/German Academic Exchange Service (DAAD).



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
