# Peer review of "Tropical drought risk: estimates combining gridded vulnerability and hazard data"

_Natural Hazards and Earth System Sciences, 2020_

## Referee Comment (RC1) · Anonymous Referee #1 · 6 Dec 2020

This manuscript describes the evaluation of drought risk in four tropical catchments of Brazil, Colombia, Costa Rica and Cambodia/Vietnam.

Drought in tropical areas has been scarcely evaluated; therefore, this paper makes a substantial contribution to the assessment of drought considering hazard and vulnerability to achieve a compound drought risk index.

Overall, the manuscript is clear and well written; it is concise and easy to follow. The tables and figures are descriptive enough to communicate the methods and results.

The Introduction deals with the main aspect to place the study in context and contains relevant and updated citations to other published works. A minor observation is in line 69, where I suggest using "livestock" instead of "cattle" grazing, as livestock is a

more general word to include the production of domestic animals, while cattle refers only to large rumiant animals, mainly with horns. Unless cattle are the only type of domestic animals for production in the four catchments, "livestock" seems to be a more appropriate term.

In general, the methodology is scientifically sounded, but I have some comments: The assessment of drought hazard seems to be straight forward. However, it would be useful if authors provide the time periods of analyses for each studied catchments, then the reader can have an idea of whether the periods are long enough to reflect the variability of the precipitation. It is not clear if they used the same period from all the catchments. Related to this, in line 144, authors mention they used a period from 1980-2018 to analyze CHIRPS data, but in line 156, they say they considered a period from 1980-2017.

Why did the authors only use the vegetation condition index to assess the vegetation condition within the drought hazard component? Why didn't use the Vegetation Health Index (VHI) which integrates the temperature condition (which can also be computed from MODIS data) in addition to vegetation index?

Vulnerability was computed using different freely available data sources which have different spatial resolutions. I suggest including these spatial resolutions in Table 2 to provide more information regarding these different data sources. Combining different sources with different spatial resolutions always represent a challenge for any spatial analysis. I am aware that authors used the best available spatial datasets regarding livestock, cropland and population densities, as well as data on roads and GDP. However, authors should consider adding some lines in their discussion regarding the way these different spatial resolutions may affect their results. I am aware that authors make a discussion regarding that more detailed data would be necessary to obtain better results at a more local scale (but may not be available or existent). Although they state in lines 218/219 that all grids were resampled to a common cell size, this does not mean that all grid layers contain the same level of detail. Thus, my suggestion is to discuss

briefly about the implications of integrating different scales on their results.

There are some errors of omission of reference (e. g. lines 192, and 201/202 and 231). I guess they may be caused during the conversion of the original file into a PDF file. Please check and amend.

The results section contains enough description of the main findings and all maps, tables and figures are adequate.

In the Discussion section, authors start saying that they get "plausible" results. I suggest removing "plausible", as it seems that they were comparing their results to real data for validation, which is not the case. Authors did this only for the Muriaé catchment; therefore, I suggest following the same style they used to report results for the other catchments (in which they did not use "plausible").

Although the Discussion is rich in the way the authors analyze and interpret their results, I found a lack of references to put their study in a more general context. For example, how do this approach to the evaluation of drought risk is similar to other studies? Does this study have advantages or disadvantages compared to other drought risk studies? This would outline the utility of the approach presented in the manuscript.

I hope the authors and editor find my comments useful.

---

## Referee Comment (RC2) · Anonymous Referee #2 · 1 Jan 2021

Review of "Tropical drought risk: estimates combining gridded vulnerability and hazard data", Nauditt et al.

This manuscript proposes a method to map drought risk in small to medium sized tropical basins, using mainly globally available gridded datasets. The objectives the paper sets out to are of interest, and the need for well-developed methods to assess drought risk to support water management decisions is clear. However, despite these objectives and needs, I feel the scientific merit and method development presented to be weak, with many of the results appearing to be somewhat trivial. It is also somewhat unclear what the role is of the four cases presented. On the one hand the paper presents a method to assess risk, and then presents these results of the method applied to each of these cases. However, the paper does not explore the strengths and

weaknesses of the proposed method, critically reflecting on limitations or in places simplistic assumptions, and does not employ the four cases to underpin such a critical assessment through for example a comparative assessment. Such comparative assessment that goes beyond an enumeration of the results could add some merit to the paper.

I have several concerns.

First the concept of risk that the authors present is somewhat confusing. In line 54 the concept of risk; constituting hazard, exposure and vulnerability is presented. That concept I agree to, and also aligns to the concept of risk commonly used in drought (and flood) risk assessments. See for example the recent World Bank guidance on drought risk assessment: https://reliefweb.int/report/world/assessing-drought-hazard-and-risk-principles-and-implementation-guidance. In the rest of the paper it would, however, seem that exposure and vulnerability are used interchangeably. Indeed there is some discussion on this around line 100 of the paper, but I fail to understand how in the context of the method presented these can be simply interchanged. I will discuss this later when exploring some of the characteristics of the indicator.

Another inconsistency is the importance of infrastructure, which is introduced as an important contributor to drought risk. In the abstract it is noted that this is related to water infrastructure. However, in other parts of the paper it would appear that this is road infrastructure (see Table 2). The source of the data is also unclear. In table 2 it is noted to be the CIESIN data, but this is the gridded population of the world dataset, which to the best of my understanding does not contain data on infrastructure. Also the data has a resolution of 30-arc seconds, which is about 1 km at the equator. This raises several issues on scale, as the classification classes proposed in Table 4 suggests several categories at scales lower than 1 km, which if the scale of the data used is on the order of $\sim$ 1 km means that there is insufficient resolution to support such a detailed classification. The resulting map of the Upper Magdalena basin, where only major roads seem to be considered, shows that this leads to a resulting map

of the contribution to vulnerability that is either very low, or very high. The scale of the data used for the Tempisque is, however, quite different, and there seems to be a very high infrastructural density. What is curious though is that the population maps of the Magdalena shows that the City of Bogotá is located within the basin (which is confirmed by the coordinates), which has a dense road network commensurate with a major city of ∼10 Million inhabitants. The comparison of these two maps suggests there are some major scale issues in the underlying data and what these represent. This would raise some major questions on what the overall index represents and how this can be compared between basins.

I also have some major issues with the structure of the index, which I think has major flaws. I will consider first the vulnerability index and then the hazard index. For the vulnerability index five factors are considered. However, several of these would appear to be highly correlated. For example population density and GDP. The maps for the Magdalena show these to align almost perfectly. Also the density of infrastructure is closely correlated, as it is somewhat trivial that there are more roads in densely populated cities. This means that thee of the five factors considered to contribute equally may well have a very high correlation, and thus dominate the result. In several analyses of vulnerability indices, techniques such as PCA may be applied to reduce the dimension of the variables considered, which may well be useful here. The authors also note in the paper that they find a central tendency of the index, with little evidence of severity category 1 or 5. This is I think primarily due to the trivial nature of the indicator. A simple thought experiment illustrates this. Imagine a basin with a pristine forest area, untouched by humans without any infrastructure or crops or livestock. This would result applying table 5 in values of 1,1,1,5,5 respectively, and therefore a value of the vulnerability indicator of 2.6. So what does that mean? A fully natural area has a vulnerability of 2.6? I agree that this may well be due to the equal weighting of categories, which immediately raises the question of why that choice was then made. The resulting maps show there is indeed little resolution to the index. It is also not clear in Table 4 how the classes chosen are motivated, and indeed validated. These seem

to be somewhat arbitrary.

Similar doubts can be raised for the hazard indicator. The authors note that the widely applied assessment of drought at monthly time scales is flawed for tropical catchments. One reason for this is given in Line 70: A (sic) few days without rainfall might lead to a severe precipitation deficit that can affect cattle grazing and rain-fed agricultural. I find this somewhat suggestive, and it is not further substantiated. I agree that for some crops the occurrence of e.g. dry spells, which is the more commonly used term in literature of a sequence of dry days during the wet season, may have significant impact on yield. But generally this would be more than just a few days, often a dry spell is considered in excess of 5 days, or sometimes 10. I would argue that this would depend very much on the crop, and local conditions such as soils. In the hazard indicator, a division is then made of a short duration and a long duration event, which indeed also considers longer periods; so how does that reflect back on the argument of the need for an assessment at daily scale. However, it would appear to me that the selection of the length of period is somewhat arbitrary, and the same thresholds are applied for all four cases. I would think this should depend somewhat on the variability of the climate? I think it would be good to explore the distributional properties of a climate. I would expect the distribution of the climate of the Tempisque and to be quite different than the Upper Magdalena, with the latter having a much lower coefficient of variation. The same holds I am sure for the other catchments though I am myself less familiar with their climate regimes. A similar discussion can be extended to the hydrological indicators. These are considered across different periods to the meteorological indicators, but again choices made seem somewhat arbitrary. There is no consideration of autocorrelations, which for discharges during low flow periods would be expected to be quite high, in particular in large basins such as the Upper Magdalena, and lower in small basins such as the Tempisque. Given these strong autocorrelations, it is unclear to me if there may be some form of double-counting (or are all short duration events that coincide with a long duration event removed?). All these details on the construction of the index would need to be clarified.

It is also unclear to me how anthropogenic influences are taken into account. If I understand correctly, the hydrostreamer approach used distributes the hydrological outputs of a global hydrological model given the temporal resolution of a gauge, which may be influenced by the operation of a hydropower station. Does this then translate to the same distribution (temporally) upstream of the gauge, and therefore perhaps upstream of the reservoir? That is not clear to me, and raises questions to how representative that then is of upstream drought hazard? Also the index does not consider temperature (evaporation), which in the drier basins may have an important impact.

Other remarks on the method are logically on the equal weighting of the constituent parts of the indicator. The sensitivity of these weights is not explored anywhere in the paper. I realise that the authors suggest that in all four basins local experts have corroborated the results. However, I do think that it is very unclear what that corroboration actually constitutes. Was some methodological approach chosen to validate results found? What benchmarks were used? Were local data on e.g. impacts used? I also do have many much more detailed remarks, where there are minor flaws in writing, style and presentation. Units are not always correct (check Table 4, cropland and population columns), and at times quite suggestive claims are made. For example, in line 135 the authors claim that: Available discharge observations data in the study regions (Figure 1) do not allow 135 to display the spatial variability in hydrological behaviour. However, in the Upper Magdalena they report to have 46 stations in a 49382 km2 basin. This translates to a density of one station per 1000-2000 km2. I would argue this is very reasonable, if not even reasonably high. The Muriaé has a similar density, it is a little lower for the Tempisque and indeed much lower for the Srepok. There are many other such remarks that are made by the authors that seem somewhat suggestive.

Concluding, I think at face value the paper seems to present an interesting analysis, but when digging a little deeper there are many methodological issues, and in my opinion raises more questions than it answers. My recommendation would therefore be to not consider this suitable for publication in its current form as it lacks a well-developed

scientific analysis.

---

## Author Comment (AC1) · 15 Feb 2021

**Response to Reviewer # 1**

Dear anonymous Reviewer (#1),

Thank you for taking the time to review our manuscript and for your valuable comments! In the following sections, we respond to your helpful suggestions and concerns:

> ***This manuscript describes the evaluation of drought risk in four tropical catchments of Brazil, Colombia, Costa Rica and Cambodia/Vietnam. Drought in tropical areas has been scarcely evaluated; therefore, this paper makes a substantial contribution to the assessment of drought considering hazard and vulnerability to achieve a compound drought risk index. Overall, the manuscript is clear and well written; it is concise and easy to follow. The tables and figures are descriptive enough to communicate the methods and results.***

Thank you! We appreciate that you recognize the adequate structure and general content of our manuscript!

> ***The Introduction deals with the main aspect to place the study in context and contains relevant and updated citations to other published works. A minor observation is in line 69, where I suggest using "livestock" instead of "cattle" grazing, as livestock is a more general word to include the production of domestic animals, while cattle refers only to large rumiant animals, mainly with horns. Unless cattle are the only type of domestic animals for production in the four catchments, "livestock" seems to be a more appropriate term.***

We agree with your comment; and would replace the term "cattle" by "livestock".

> ***In general, the methodology is scientifically sounded, but I have some comments: The assessment of drought hazard seems to be straight forward. However, it would be useful if authors provide the time periods of analyses for each studied catchments, then the reader can have an idea of whether the periods are long enough to reflect the variability of the precipitation. It is not clear if they used the same period from all the catchments. Related to this, in line 144, authors mention they used a period from 1980-2018 to analyze CHIRPS data, but in line 156, they say they considered a period from 1980-2017.***

Thank you for this important hint. For the SPI calculation, we used the Chirps v2.0 precipitation data set with a common period from 1981-2018 for each of the four study catchments. We will correct the wrongly mentioned data period (1980-2017).

> ***Why did the authors only use the vegetation condition index to assess the vegetation condition within the drought hazard component? Why didn´t you use the Vegetation Health Index (VHI) which integrates the temperature condition (which can also be computed from MODIS data) in addition to vegetation index?***

We appreciate this comment and agree with Reviewer 1 that VHI has been widely applied in drought related research. VHI combines the Vegetation Condition Index (VCI) and the

Temperature Condition Index (TCI) that relates to evaporative demand. The validity of the VHI as drought assessment index is based on the assumption that the land surface temperature (LST) and the normalized difference vegetation index (NDVI) inversely vary over time (i.e., there is a negative NDVI-LST correlation) (Karnieli et al., 2010). Under drought periods, LST is expected to be high, while NDVI is expected to be low. However, some studies have shown that i) this negative correlation is not necessarily applicable everywhere (e.g., Smith and Choudhury 1991; Karnieli et al., 2006; Sun and Kafatos, 2007); and ii) there are often positive NDVI-LST correlations over the tropics and high latitudes, which represent areas where vegetation growth is energy limited (Nemani et al. 2003; Julien and Sobrino 2009). In humid regions (such as the tropics), increased temperatures may lead to increased plant biomass and height as confirmed by several studies through warming experiments (Van Wijk et al. 2003; Stow et al. 2004; Walker et al. 2006). We therefore decided to use VCI, which we consider suitable to detect vegetation related drought anomalies in the tropics, especially in tropical grassland ecosystems. We would include this explanation in a potential update of the manuscript.

> ***Vulnerability was computed using different freely available data sources which have different spatial resolutions. I suggest including these spatial resolutions in Table 2 to provide more information regarding these different data sources. Combining different sources with different spatial resolutions always represent a challenge for any spatial analysis. I am aware that authors used the best available spatial datasets regarding livestock, cropland and population densities, as well as data on roads and GDP. However, authors should consider adding some lines in their discussion regarding the way these different spatial resolutions may affect their results. I am aware that authors make a discussion regarding that more detailed data would be necessary to obtain better results at a more local scale (but may not be available or existent). Although they state in lines 218/219 that all grids were resampled to a common cell size, this does not mean that all grid layers contain the same level of detail. Thus, my suggestion is to discuss briefly about the implications of integrating different scales on their results.***

Thank you for this valuable hint, this information is indeed missing! We will include the spatial resolution information in Table 2 and incorporate a section in the discussion related to the influence of differing spatial resolutions and the expected implications of increasing spatial resolution on the results.

**Table 2: Vulnerability related data sets**

| Population | Density | GHS Population Grid 2015 | 250m |
|---|---|---|---|
| Livestock | Density | Gridded Livestock of the World (GLW) | 5km |
| Crop area | Density | Global Agricultural Lands 2000 | 1km |
| Proximity to Infrastructure | Euclidean distance | Major roads | 30m |
| GDP PPP | GDP per capita | https://doi.org/10.5061/dryad.dk1j0 | 1km |

The data sets were resampled to the highest resolution: 30m. This lowers the accuracy of the vulnerability information in the grid cells for the resampled data products, by transferring

the same value of the lower resolution cells to all cells lying within the larger cell. This is especially relevant for livestock and crop density that showed more spatial variation. We will discuss the implications of resampling to a higher spatial resolution for each indicator in section 4.3.

> ***The results section contains enough description of the main findings and all maps, tables and figures are adequate.***

Thank you for this positive remark!

> ***In the Discussion section, authors start saying that they get "plausible" results. I suggest removing "plausible", as it seems that they were comparing their results to real data for validation, which is not the case. Authors did this only for the Muriaé catchment; therefore, I suggest following the same style they used to report results for the other catchments (in which they did not use "plausible").***

Thank you for this recommendation. We will remove the term "plausible" and change the sentence accordingly: "Our results were able to display drought risk hotpots in the Muriaé basin…"

> ***Although the Discussion is rich in the way the authors analyse and interpret their results, I found a lack of references to put their study in a more general context. For example, how is this approach to the evaluation of drought risk similar to other studies? Does this study have advantages or disadvantages compared to other drought risk studies? This would outline the utility of the approach presented in the manuscript.***

We agree that such a discussion would add more value to the manuscript. We would include a separate section on the strengths and weaknesses of our approach in the context of the wider literature.

*Section 4.4: Drought risk assessment*
Multiple assessment approaches have been developed to evaluate drought risk in regions worldwide; each of them strongly depending on site-specific drought hazard and vulnerability characteristics (Hall and Leng, 2019; Hagenlocher et al., 2019).
Vogt et al. (2018) distinguished between two main drought risk assessment classes: the "outcome or impact approach" and the "contextual or factor approach" (Brooks et al., 2003; González-Tánago et al., 2016, Naumann et al., 2018). The "outcome or impact approach" is based on quantitative data about historical impacts as proxies for vulnerability, and focusses on the expected losses due to a particular hazard (eg. Kumar and Panu, 1997; Stagge et al., 2015; Bachmair et al., 2015; Blauhut et al., 2015&2016). Regression models are used to determine the probability or "likelihood" of occurrence of an impact to occur in dependence of a drought hazard anomaly.
The "contextual or factor approach" combines hazard with exposed social and economic factors or assets as vulnerability proxies. The approach uses normalized and weighted

indicators/indices that describe the relative susceptibility of a location to damaging effects of a drought (e.g. Naumann et al., 2014&2019; Carrão et al., 2016).

Ideally both approaches should be combined; shortcomings – related to eg. limited data and their spatio-temporal resolution or arbitrary decisions related to weighting and data masking - - are discussed by the authors mentioned above and in Brooks et al. (2003), Vogt et al. (2018) and Hagenlocher et al. (2019).

Our study belongs to the "contextual or factor approach" category and suggests indicators to evaluate tropical drought risk under data scarcity. Rather than providing a generic approach, it evaluates the suitability of each individual index and gridded data set to be applied to four different tropical study regions. It attempts to address spatial variation of hazard in response to catchment characteristics and shorter but damaging tropical drought hazards.

**_I hope the authors and editor find my comments useful._**

Yes! Thank you for this constructive feedback!

**References**

Bachmair, S., Svensson, C., Hannaford, J., Barker, L. J., and Stahl, K.: A quantitative analysis to objectively appraise drought indicators and model drought impacts, Hydrol. Earth Syst. Sci., 20, 2589–2609, https://doi.org/10.5194/hess-20-2589-2016, 2016.

Blauhut, V., Stahl, K., Stagge, J. H., Tallaksen, L. M., Stefano, L. de, and Vogt, J.: Estimating drought risk across Europe from reported drought impacts, drought indices, and vulnerability factors, Hydrol. Earth Syst. Sci., 20, 2779–2800, https://doi.org/10.5194/hess-20-2779-2016, 2016.

Brooks, N., 2003. Vulnerability, Risk and Adaptation: A Conceptual Framework. Working Paper 38, Tyndall Centre for Climate Change Research, University of East Anglia, Norwich. https://gsdrc.org/document-library/vulnerability-risk-and-adaptation-a-conceptual-framework/

González-Tánago, I., Urquijo, J., Blauhut, V., Villarroya, F. and De Stefano, L.: Learning from experience: a systematic review of assessments of vulnerability to drought, Nat. Hazards, 80, 951–973, doi:10.1007/s11069-015-2006-1, 2016.

Hagenlocher, M., Meza, I., Anderson, C. C., Min, A., Renaud, F. G., Walz, Y., Siebert, S., and Sebesvari, Z.: Drought vulnerability and risk assessments: state of the art, persistent gaps, and research agenda, Environ. Res. Lett., 14, 83002, https://doi.org/10.1088/1748-9326/ab225d, 2019.

Hall, JW. Leng, G., 2019. Can we calculate drought risk… and do we need to? Wiley WIRE Water, https://doi.org/10.1002/wat2.1349

Julien, Y., and J. A. Sobrino, 2009: The Yearly Land Cover Dynamics (YLCD) method: An analysis of global vegetation from NDVI and LST parameters. Remote Sens. Environ., 113, 329–334.

Karnieli, A., Agam, N., Pinker, R.T., Anderson, M., Imhoff, M.L., Gutman, G.G., Panov, N. and Goldberg, A., 2010. Use of NDVI and land surface temperature for drought assessment: Merits and limitations. Journal of climate, 23(3), pp.618-633.

Karnieli, A., M. Bayasgalan, Y. Bayarjargal, N. Agam, S. Khudulmur, and C. J. Tucker, 2006: Comments on the use of the vegetation health index over Mongolia. Int. J. Remote Sens., 27, 2017– 2024.

Kumar and V., Panu, U., 1997. Predictive assessment of severity of agricultural droughts based on agro-climatic factors. J. Am. Water Resour. Assoc. 33 (6), 1255–1264.

Naumann, G., Barbosa, P., Garrote, L., Iglesias, A., Vogt, J.: Exploring drought vulnerability in Africa: an indicator based analysis to be used in early warning systems. Hydrol. Earth Syst. Sci. 18, 1591–1604, 2014.

Naumann, G., Vargas, W., Barbosa, P., Blauhut, V., Spinoni, J., and Vogt, J.: Dynamics of Socioeconomic Exposure, Vulnerability and Impacts of Recent Droughts in Argentina, Geosciences, 9, 39, https://doi.org/10.3390/geosciences9010039, 2019.Nemani, R., and S. Running, 1989: Estimation of regional surface resistance to evapotranspiration from NDVI and thermal-IR AVHRR data. J. Appl. Meteor., 28, 276–284.

Smith, R. C. G., and B. J. Choudhury, 1991: Analysis of normalized difference and surface temperature observations over southeastern Australia. Int. J. Remote Sens., 12, 2021–2044.

Stow, D. A., and Coauthors, 2004: Remote sensing of vegetation and land-cover change in Arctic tundra ecosystems. Remote Sens. Environ., 89, 281–308.

Sun, D., and M. Kafatos, 2007: Note on the NDVI-LST relationship and the use of temperature-related drought indices over North America. Geophys. Res. Lett., 34, L24406, doi:10.1029/ 2007GL031485.

Van Wijk, M. T., M. Williams, J. A. Laundre, and G. R. Shaver, 2003: Interannual variability of plant phenology in tussock tundra: Modelling interactions of plant productivity, plant phenology, snowmelt and soil thaw. Global Change Biol., 9, 743–758.

Vogt, J., Naumann, G., Masante, D., Spinoni, J., and Barbosa, P.: Drought Risk Assessment and Management. A Conceptual Framework, https://doi.org/10.2760/919458, available at: https://www.researchgate.net/publication/329451050_Drought_Risk_Assessment_and_Manag ement_A_Conceptual_Framework, 2018.

Walker, M. D., and Coauthors, 2006: Plant community responses to experimental warming across the tundra biome. Proc. Natl. Acad. Sci. USA, 103, 1342–1346.

---

## Author Comment (AC2) · 15 Feb 2021

**Response to Reviewer # 2**

Dear anonymous Reviewer (#2),
Thank you for reviewing our manuscript and for your valuable comments.

In the following sections, we will answer to your suggestions and concerns:

1. ***This manuscript proposes a method to map drought risk in small to medium sized tropical basins, using mainly globally available gridded datasets. The objectives the paper sets out to are of interest, and the need for well-developed methods to assess drought risk to support water management decisions is clear.***

Thank you for acknowledging our objectives and for recognizing the research demand.

2. ***However, despite these objectives and needs, I feel the scientific merit and method development presented to be weak, with many of the results appearing to be somewhat trivial. It is also somewhat unclear what the role is of the four cases presented. On the one hand the paper presents a method to assess risk, and then presents these results of the method applied to each of these cases. However, the paper does not explore the strengths and weaknesses of the proposed method, critically reflecting on limitations or in places simplistic assumptions, and does not employ the four cases to underpin such a critical assessment through for example a comparative assessment. Such comparative assessment that goes beyond an enumeration of the results could add some merit to the paper.***

Thank you for this recommendation. We agree that a section with a more detailed comparative assessment of the index performance in the differing tropical study regions would nicely complement our manuscript. We tried to provide a comparative and critical interpretation of the individual index results in sections 4.2 and 4.3. If accepted for publication, we will include a qualitative and where possible a quantitative comparative evaluation to the manuscript, that contrasts the results for factors such as size and spatial scale, topography, geology, hydro-climatic characteristics and related seasonality and variability, land cover, differing land uses and level of anthropogenic influence, highlighting strengths and weaknesses of our approach.
We do not claim that our methodology, the selected data sets, suggested thresholds, indices and classifications are universally transferable to any region. We rather attempted to "test" these in different tropical climates and environments that experienced severe drought impacts in recent years.

3. ***I have several concerns. First the concept of risk that the authors present is somewhat confusing. In line 54 the concept of risk; constituting hazard, exposure and vulnerability is presented. That concept I agree to, and also aligns to the concept of risk commonly used in drought (and flood) risk assessments. See for example the recent World Bank guidance on drought risk***

*assessment: [https://reliefweb.int/report/world/assessing-drought-hazard-and-riskprinciples-and-implementation-guidance](https://reliefweb.int/report/world/assessing-drought-hazard-and-riskprinciples-and-implementation-guidance).*
*In the rest of the paper it would, however, seem that exposure and vulnerability are used interchangeably. Indeed there is some discussion on this around line 100 of the paper, but I fail to understand how in the context of the method presented these can be simply interchanged. I will discuss this later when exploring some of the characteristics of the indicator.*

Thank you for this observation and for sharing the interesting World Bank drought risk report. In a revised version of our manuscript, we could certainly improve the part on how we define vulnerability. We could explain this in more detail in the Introduction and in the Methods´ section (2.3).

We are aware that drought vulnerability and exposure address two different concepts. However, due to missing information on drought impacts, sensitivity, adaptive capacity or other potential constituents of vulnerability, we were only able to use exposure data as proxies for vulnerability. Our approach corresponds to the concept of IPCC 2014, where vulnerability is described as a combination of exposure, sensitivity and adaptive capacity, while risk is understood as a combination of vulnerability and hazard (IPCC, 2014).

Drought risk concepts and related terms are still not universally manifested as they are in more traditional disciplines (Brooks et al., 2003). They are used differently in a large variety of studies, each suggesting own risk definitions and assessment methods, indices and data sets. Vulnerability definitions and concepts are even more diverse (e.g. De Stefano et al., 2015; Gonzáles-Tánago et al., 2015; Naumann et al., 2018; Hagenlocher et al., 2019).

4. *Another inconsistency is the importance of infrastructure, which is introduced as an important contributor to drought risk. In the abstract it is noted that this is related to water infrastructure. However, in other parts of the paper it would appear that this is road infrastructure (see Table 2). The source of the data is also unclear. In table 2 it is noted to be the CIESIN data, but this is the gridded population of the world dataset, which to the best of my understanding does not contain data on infrastructure.*

Thank you for sharing your concern about the infrastructure related vulnerability. Maybe our formulation was misleading. Although we think that water infrastructure is a relevant indicator for vulnerability, we did not introduce infrastructure as an "important contributor to drought risk"; the "lack of water infrastructure" is only mentioned in the abstract as one of the reasons why the adaptive capacity in many tropical water abundant regions is low.

Based on our a priori review of available data sets, we identified the CIESIN Global Roads map as one of the five gridded and open-access vulnerability related data sets that was available for all of the four study regions (Global Roads Open Access Dataset (gROADSv1), http://sedac.ciesin.columbia.edu/data/set/groads-global-roads-open-access-v1).

Water infrastructure (e.g. Aquastat) datasets as used in Carrão et al. (2016) and Naumann et al. (2015, 2018&2019) are not available for all of the four study regions.

To increase the value of our manuscript for future research and stakeholder applications, we

could suggest additional gridded data sets that are only available for individual basins or that were published more recently. Our scripts would easily allow to mask and interpret other gridded data sets to describe site specific drought vulnerability. We could include such a table with data sources per study region at the end of the section where we compare the index performance in each of the basins (e.g. 4.4).

Upon publication, we will publish the developed scripts on GitHub for open access.

> 5. ***This raises several issues on scale, as the classification classes proposed in Table 4 suggests several categories at scales lower than 1 km, which if the scale of the data used is on the order of _ 1 km means that there is insufficient resolution to support such a detailed classification. The resulting map of the Upper Magdalena basin, where only major roads seem to be considered, shows that this leads to a resulting map of the contribution to vulnerability that is either very low, or very high. The scale of the data used for the Tempisque is, however, quite different, and there seems to be a very high infrastructural density. What is curious though is that the population maps of the Magdalena shows that the City of Bogotá is located within the basin (which is confirmed by the coordinates), which has a dense road network commensurate with a major city of _10 Million inhabitants. The comparison of these two maps suggests there are some major scale issues in the underlying data and what these represent. This would raise some major questions on what the overall index represents and how this can be compared between basins.***

Thank you for sharing your concern about the differences in the infrastructure maps. The fact that for the Upper Magdalena only major roads are displayed might be associated with the much larger catchment area compared to eg. the Tempisque basin. Also, the steep topography and the remote, densely forested National Park in the West of the Magdalena basin does not allow much infrastructure. The Tempisque accounts for a dense road network for beach tourism and intensive agriculture.

However, we agree that the difference between the infrastructure map of the Tempisque compared to the ones for the other basins is too strong and there might be an error in the result. In case we are invited to submit a revised manuscript, we will carefully check the raw data, script calculations and classification again before reproducing the map.

> 6. ***I also have some major issues with the structure of the index, which I think has major flaws. I will consider first the vulnerability index and then the hazard index. For the vulnerability index five factors are considered. However, several of these would appear to be highly correlated. For example population density and GDP. The maps for the Magdalena show these to align almost perfectly. Also the density of infrastructure is closely correlated, as it is somewhat trivial that there are more roads in densely populated cities. This means that thee of the five factors considered to contribute equally may well have a very high correlation, and thus dominate the result. In several analyses of vulnerability indices, techniques such as PCA may be applied to reduce the dimension of the variables considered, which may well be useful here. The authors also note in the paper that they find a central tendency of the index, with little evidence of severity category 1 or 5. This is I think primarily due to the trivial nature of the indicator. A simple thought experiment illustrates this. Imagine a basin with a pristine forest area, untouched by humans without any infrastructure or***

*crops or livestock. This would result applying table 5 in values of 1,1,1,5,5 respectively, and therefore a value of the vulnerability indicator of 2.6. So what does that mean? A fully natural area has a vulnerability of 2.6? I agree that this may well be due to the equal weighting of categories, which immediately raises the question of why that choice was then made. The resulting maps show there is indeed little resolution to the index. It is also not clear in Table 4 how the classes chosen are motivated, and indeed validated. These seem to be somewhat arbitrary.*

We understand that the index choice and weighting system applied can appear arbitrary. The pristine basin example you mention is excellent. We are aware that an adequate vulnerability assessment would require more contrasting proxies – as also shown in the scientific articles mentioned above and in our manuscript. For the forested areas like much of the Srepok, our weighted vulnerability index showed the lowest vulnerability compared to the agricultural regions, but is also not necessarily the ideal combination. The main purpose was to assess the adequacy of the available data sets to describe vulnerability in differing tropical study regions, which we would emphasize much more in a potentially revised manuscript. Eg. for rural areas, proximity to road infrastructure is a relevant indicator, while it is not in an urban area (as you say, it is likely to be negatively correlated with population density).

Although GDP per capita (PPP) and grid cell (Kummu et al., 2018) is likely to be high in urban and hence more developed areas, it is not necessarily correlated with population density. As we show in Figure 4 and explain under section 4.3 in line 444, the Population Density index in general identified very few pixels as being vulnerable. In the context of our study, we therefore found it to be useful to account for differences in GDP per capita in rural areas.

We agree that applying cross correlation and PCA and resulting data masking would be useful tools to filter relevant vulnerability information and indices. Regrettably, the low number of proxies does not offer many values to choose from. After weighting, the severity values 1 and 5 would only be a result, if all five indices were showing values of either 1 or 5. We argue that one indicator resulting in 5 alone is not enough to show highest vulnerability and that in relative terms, differences among grid cells have been identified.

Our choice was made to facilitate the interpretation of the results and to make these results (also negative ones) more transparent. We tried to describe these issues in the discussion and conclusion, structured by addressing individual study regions, hazard and vulnerability indices; but we will make this clearer or even provide a table that shows our derived recommendations.

We will explore spatial cross correlations between the indices and based on the results try to adapt and justify our weights. Other approaches would require independent validation, which the available information and data do not allow.

7. *Similar doubts can be raised for the hazard indicator. The authors note that the widely applied assessment of drought at monthly time scales is flawed for tropical catchments. One reason for this is given in Line 70: A (sic) few days without rainfall might lead to a severe precipitation deficit that can affect cattle grazing and rain-fed agricultural. I find this somewhat suggestive, and it is not further substantiated. I agree that for some crops the occurrence of e.g. dry spells, which is the more commonly used term in literature of a sequence of dry days during the wet season, may have significant impact on yield. But generally this would be more than just a few days, often a dry spell is considered in excess of 5 days, or sometimes 10. I would argue that this would depend very much on the crop, and local conditions such as soils. In the hazard indicator, a division is then made of a short duration and a long duration event, which indeed also considers longer periods; so how does that reflect back on the argument of the need for an assessment at daily scale. However, it would appear to me that the selection of the length of period is somewhat arbitrary, and the same thresholds are applied for all four cases. I would think this should depend somewhat on the variability of the climate? I think it would be good to explore the distributional properties of a climate. I would expect the distribution of the climate of the Tempisque and to be quite different than the Upper Magdalena, with the latter having a much lower coefficient of variation. The same holds I am sure for the other catchments though I am myself less familiar with their climate regimes.*

We thank you for having a detailed look at our hazard evaluation. The hazard indices were developed for regions where livestock/milk production (grassland) and/or rainfed agriculture is dominating (eg. the Northern part of upper Magdalena and the Muriae). These sectors and regions have been strongly affected by frequent short-term droughts in recent years. We selected the duration of short hydrological droughts based on these experiences.  In the downstream part of the Upper Magdalena, for instance, it usually rains every day of the year. Monthly indices would exclude shorter events that follow or are followed by strong precipitation events within the same month, which often occurs in tropical regions. The number of events identified gives an indication on how relevant the indices are.

As for the vulnerability proxies, the hazard indices might not be relevant for all regions. The periods and classification values can be adapted in the codes after an a-priori hydro-climatic characterization of a given tropical region. The same applies to the incorporation of sawing periods and soil characteristics to the index design. Addressing more site specific and spatially distributed characteristics would complicate our methodology and make a regional inter-comparison of index performance even more difficult.

We have conducted several of the a priori studies you suggest, but a demonstration and application to each of the study regions would go beyond of the scope of this article, which has the aim to evaluate indicators and indices for data scarce tropical regions and provide a reasonable spatial and temporal resolution to support local decision-making.

We have also tried to highlight this in the sections of the discussion and in the conclusion, but agree that the shortcomings and recommendations could be evaluated in a more systematic way (e.g. in a table) in a potential new version of the manuscript.

8. *A similar discussion can be extended to the hydrological indicators. These are considered across different periods to the meteorological indicators, but again choices made seem somewhat arbitrary. There is no consideration of autocorrelations, which for discharges during low flow periods would be expected to be quite high, in particular in large basins such as the Upper Magdalena, and lower in small basins such as the Tempisque. Given these strong autocorrelations, it is unclear to me if there may be some form of double-counting (or are all short duration events that coincide with a long duration event removed?). All these details on the construction of the index would need to be clarified.*

We apologize if this has not been made clear enough. As shown in Table 3, the duration of short drought events has an upper limit and therefore there is no overlapping of events. Given that the vulnerability index accounts for five indicators, we decided to give reasonable weights to hazard. Also, we wanted to take into account long term droughts.
As you suggest under section 6., we can apply cross-correlation to identify and potentially exclude overlapping and outweighing factors in general terms.

*It is also unclear to me how anthropogenic influences are taken into account. If I understand correctly, the hydrostreamer approach used distributes the hydrological outputs of a global hydrological model given the temporal resolution of a gauge, which may be influenced by the operation of a hydropower station. Does this then translate to the same distribution (temporally) upstream of the gauge, and therefore perhaps upstream of the reservoir? That is not clear to me, and raises questions to how representative that then is of upstream drought hazard?*

We agree that human modifications of streamflow like hydropower operation can be a major problem to simulate daily streamflow with Hydrostreamer. In our study, we only faced problems in the Magdalena basin (described in line 418 under section 4.2). These might have occurred also due to inconsistencies in the IDEAM discharge data that were used to validate Hydrostreamer. A further problem for any hydro-climatic application in the Magdalena is the unusually steep and heterogeneous topography. We are working on a more refined data set using corrected and modelled data.
Concerning the application of our daily hydrological index in anthropogenically impacted catchments, we think that the advantage of a hydrological drought index is that it can take into account both, "natural" upstream and human impacted drought anomalies (water scarcity). This also implies that the results of a hydrological drought index are not necessarily overlapping with meteorological droughts.

*Also the index does not consider temperature (evaporation), which in the drier basins may have an important impact.*

We agree that in some parts of the study regions (and seasons), like the most upper part of the Magdalena (eg. the Huila Department) evapotranspiration and temperature would be

valuable additional variables to detect moisture anomalies. For the wetter tropical regions, we have found that temperature has only a minor influence on drought index performance (Vicente-Serrano et al., 2009). Generally speaking, we expect that the hydrological indices are adequate to show the overall hazard anomaly of a catchment system. In an updated manuscript, we can suggest to apply other indices that incorporate temperature like SPEI, VHI, TCI for drier regions or regions with a long dry season. We will also suggest that for regions with dry seasons lasting between 3 and 8 months, an SPEI adapted to the median dry season start, duration and end would be a valuable complementary index.

> ***Other remarks on the method are logically on the equal weighting of the constituent parts of the indicator. The sensitivity of these weights is not explored anywhere in the paper.***

As you proposed, we suggest to include a cross correlation analysis to a potentially updated manuscript. It will allow to derive weights and perform an uncertainty analysis to see how changes in weights impact the results.

> ***I realise that the authors suggest that in all four basins local experts have corroborated the results. However, I do think that it is very unclear what that corroboration actually constitutes. Was some methodological approach chosen to validate results found? What benchmarks were used? Were local data on e.g. impacts used?***

We agree that more validation of the results could be presented and we will try to improve this in a potential new version of the manuscript. We could summarize the information provided by the representatives of local universities and stakeholders, which consisted in formal interviews with identical questions for a sound qualitative analysis. Please understand that we tried to structure the manuscript according to our key objectives and findings; also to avoid exceeding a reasonable amount of words.

> ***I also do have many much more detailed remarks, where there are minor flaws in writing, style and presentation. Units are not always correct (check Table 4, cropland and population columns), and at times quite suggestive claims are made.***

Thank you for identifying mistakes in writing, style and presentation. It would be helpful if the reviewer could indicate the lines in the manuscript associated with these "flaws".
Which unit is wrong in Table 4 (Vulnerability indicators)?

> *For example, in line 135 the authors claim that: Available discharge observations data in the study regions (Figure 1) do not allow to display the spatial variability in hydrological behaviour. However, in the Upper Magdalena they report to have 46 stations in a 49382 km2 basin. This translates to a density of one station per 1000-2000 km2. I would argue this is very reasonable, if not even reasonably high. The Muriaé has a similar density, it is a little lower for the Tempisque and indeed much lower for the Srepok. There are many other such remarks that are made by the authors that seem somewhat suggestive.*

We aimed at using a gridded high resolution hydrological product that can take into account differences in spatially varying biophysical catchment characteristics. We used IDEAM station data to validate hydrostreamer. As the title suggests, for our index analysis we only used gridded data sets.

> **9. Concluding, I think at face value the paper seems to present an interesting analysis, but when digging a little deeper there are many methodological issues, and in my opinion raises more questions than it answers. My recommendation would therefore be to not consider this suitable for publication in its current form as it lacks a well-developed scientific analysis.**

We regret that you disagree with our methodological approach, but we are confident that we can reasonably alleviate your concerns in a revised paper. We hope that our clarifications and suggestions for improvement provide convincing arguments to be given the opportunity to revise the manuscript.

**References**

Blauhut, V., Stahl, K., Stagge, J. H., Tallaksen, L. M., Stefano, L. de, and Vogt, J.: Estimating drought risk across Europe from reported drought impacts, drought indices, and vulnerability factors, Hydrol. Earth Syst. Sci., 20, 2779–2800, https://doi.org/10.5194/hess-20-2779-2016, 2016.

Carrão, H., Naumann, G., and Barbosa, P.: Mapping global patterns of drought risk: An empirical framework based on sub-national estimates of hazard, exposure and vulnerability, Global Environmental Change, 39, 108–124, https://doi.org/10.1016/j.gloenvcha.2016.04.012, 2016.

CIESIN, Columbia University Center for International Earth Science Information Network, and Information Technology Outreach Services, University of Georgia. Global Roads Open Access Data Set, version 1. NY: NASA Socioeconomic Data and Applications Center (SEDAC), https://doi.org/10.7927/H4VD6WCT (2013).

De Stefano, L., González-Tánago, I., Ballesteros, M., Urquijo, J., Blauhut, V., James, H., and Stahl, K., 2015. Methodological approach considering different factors influencing vulnerability – pan-European scale, Drought R&SPI, Technical Report no. 26, https://www.researchgate.net/publication/274536771_METHODOLOGICAL_APPROACH_CONSIDERING_DIFFERENT_FACTORS_INFLUENCING_VULNERABILITY_-_PAN-EUROPEAN_SCALE

Freire, S. & Pesaresi, M. GHS population grid, derived from GPW4, multitemporal (1975, 1990, 2000, 2015), Available at data. https://data.europa.eu/euodp/en/data/dataset/jrc-ghsl-ghs_pop_gpw4_globe_r2015a (European Commission, Joint Research Centre, 2015).

González-Tánago, I., Urquijo, J., Blauhut, V., Villarroya, F., and De Stefano, L.: Learning from experience: a systematic review of assessments of vulnerability to drought, Nat. Hazards, 80, 951–973, doi:10.1007/s11069-015-2006-1, 2015.

Hagenlocher, M., Meza, I., Anderson, C. C., Min, A., Renaud, F. G., Walz, Y., Siebert, S., and Sebesvari, Z.: Drought vulnerability and risk assessments: state of the art, persistent gaps, and research agenda, Environ. Res. Lett., 14, 83002, https://doi.org/10.1088/1748-9326/ab225d, 2019.

IPCC 2014 Climate Change 2014: Impacts, Adaptation, and Vulnerability. Part B: Regional Aspects. Contribution of Working Group II to the Fifth Assessment Report of the Intergovernmental Panel on Climate Change https://www.ipcc.ch/report/ar5/wg2/

Meza, I., Hagenlocher, M., Naumann, G., Vogt, J., and Frischen, J.: Drought vulnerability indicators for global-scale drought risk assessments: Global expert survey results report, JRC technical reports, Publications Office of the European Union, Luxembourg, 62 pp., available at: https://collections.unu.edu/eserv/UNU:7430/Meza_etal_2019_DroughtVulnerability_META.pdf, 2019.

Vogt, J., Naumann, G., Masante, D., Spinoni, J., and Barbosa, P.: Drought Risk Assessment and Management. A Conceptual Framework, https://doi.org/10.2760/919458, available at: https://www.researchgate.net/publication/329451050_Drought_Risk_Assessment_and_Management_A_Conceptual_Framework, 2018.